# Examining Routine Pediatric Vaccination Availability in Community Pharmacies in Washington State

**DOI:** 10.3390/pharmacy10060156

**Published:** 2022-11-22

**Authors:** Kimberly Caye McKeirnan, Madison Shea Motzner, Sorosh Kherghehpoush

**Affiliations:** 1Pharmacotherapy Department, College of Pharmacy and Pharmaceutical Sciences, Washington State University, Spokane, WA 99202, USA; 2St. Luke’s Health System, Boise, ID 83712, USA; 3Department of Clinical & Administrative Sciences, California Northstate University, Elk Grove, CA 95757, USA

**Keywords:** pharmacy practice, pediatric immunizations, patient education

## Abstract

To address diminishing pediatric vaccination rates resulting from the COVID-19 pandemic, the Public Readiness and Emergency Preparedness (PREP) Act allows pharmacists, technicians, and pharmacy interns to administer any vaccine that the Advisory Committee on Immunization Practices (ACIP) guidelines recommend for all patients aged 3 years and older. A survey was conducted to evaluate the role of pharmacy personnel in the community setting providing immunizations for the pediatric patients. Sixty-seven pharmacies were contacted in a state where pharmacists are allowed to administer vaccinations to any patient over the age of six months. Of the 58 respondent pharmacies offering vaccinations for pediatric patients, the most commonly reported vaccines included influenza (97%), tetanus, diphtheria, and pertussis (88%), hepatitis (71%), human papillomavirus (69%), meningococcal vaccines (66%), polio (45%), and Haemophilus influenzae type b vaccine (40%). Nearly all respondent pharmacies (56/58) reported having at least one of the ACIP-recommended routine childhood vaccinations available for patients under the age of 18. Community pharmacies are well-positioned to administer routine vaccinations to pediatric patients and address declining pediatric vaccination rates.

## 1. Introduction

Diminishing pediatric vaccination rates present a substantial concern for public health. Routine childhood vaccination series offer long-term immunity for many preventable infections and reduce community transmission. For each birth cohort in the United States receiving the ACIP-recommended vaccinations, an estimated 33,000 deaths are prevented along with 14 million cases of vaccine-preventable diseases, and direct health care costs are reduced by an estimated $9.9 billion [1]. This is in alignment with the World Health Organization (WHO) estimations of 2–3 million deaths being prevented every year due to vaccinations [2]. Evaluation of healthcare cost savings of routine childhood vaccinations for children born between 1994–2013 showed that approximately 322 million illnesses, 732,000 deaths, and $1.38 trillion in societal costs will be saved over the course of these children’s lifetime due to vaccinations [3]. Despite these successes in prevention, approximately 42,000 adults and 300 children in the United States die from vaccine-preventable diseases every year [1]. Lower vaccination rates lead to increased risk of transmission of vaccine preventable diseases, resulting in avoidable hospitalizations, death, and healthcare costs. Appendix A displays the 2022 Recommended Child and Adolescent Immunization Schedule for ages 18 years or younger for the United States [4].

The Office of Disease Prevention and Health Promotion’s goal for children receiving the recommended doses of childhood vaccine series in the United States for diphtheria, tetanus, acellular pertussis (DTaP), polio, measles, mumps, rubella (MMR), Haemophilus influenzae type b (Hib), hepatitis B, varicella, and the pneumococcal conjugate vaccine (PCV) by age 19–35 months was 80–88.7% [5]. However, nationally, the United States reached 72.8% of children receiving these vaccinations in 2020 [5]. Similarly, the World Health Organization (WHO) and the United Nations International Children’s Emergency Fund (UNICEF) reports MMR coverage has dropped worldwide from 86% in 2019 to 84% in 2020 [2]. 

Historically, most routine childhood vaccinations have been administered at pediatrician offices, but due to the COVID-19 pandemic, there have been new limitations placed on families and providers [6,7]. In the 2020 Survey of America’s Physicians: COVID-19 Impact Edition, 8% of physicians have closed their practice as a result of COVID-19 pandemic and 43% have reduced staff [6]. Similarly, 41% of physicians saw a decrease in volume of patients [6]. Given these disruptions to healthcare and decreased well-child visits, there has also been a reduction in pediatric immunization rates. One study found that from March 2020, after COVID-19 was declared a national emergency in the US, to April 2020 there was a decrease of roughly 2.5 million non-influenza vaccine doses ordered in the United States [8]. Additionally, when comparing vaccination rates from 2018 and 2019 with those seen in 2020, pediatric vaccine administration declined by as much as 71% in children ages 2 to 8 years and 13 to 17 years [9,10]. 

Fortunately, many of the ACIP-recommended childhood vaccinations are available and can be given in the community pharmacy setting by a pharmacist. To help expand community pharmacy vaccination efforts, the Department of Health and Human Services (HHS) have made several amendments and expansions to the Public Readiness and Emergency Preparedness (PREP) Act for Medical countermeasures against COVID-19 beginning March 2020. The PREP Act allows pharmacists, technicians, and pharmacy interns to administer any vaccine that the ACIP guidelines recommend for all patients aged 3 years and older and allows recently inactive healthcare personnel to administer COVID-19 vaccines [11,12]. This nationwide mandate has also expanded authority for pharmacists to vaccinate those that may have otherwise been restricted by state law requirements [13]. Additionally, many pharmacies do not require an appointment for vaccinations, and they often have longer hours of operation compared to primary care offices which may provide a more convenient avenue for patients to receive their immunizations. Companies such as CVS, Walgreens, Kroger, and Rite Aid advertise walk-in or scheduled vaccination appointments [14,15,16,17].

Recent data shows demand for pharmacist-administered vaccinations in the United States. As of February 2022, over 200 million COVID-19 vaccines have been administered in the community pharmacy setting [18]. Since the FDA approval of the Pfizer–BioNTech COVID-19 vaccine for children aged five to eleven years old, approximately 9 million children have received at least one dose of COVID-19 vaccine as of February 2022 [19] with approximately one-third of these vaccinations being administered in the pharmacy setting [20]. Along with increase demand for vaccines, states like Idaho also allow pharmacy technicians to administer vaccines, improving the ability of pharmacies to meet the demand [21,22]. 

While these data show excellent utilization of pharmacists on a national level for COVID-19 vaccination efforts, there are limited data on routine childhood vaccine administration rates. In most states, pharmacists have the authority to administer vaccines to pediatrics as young as six months old, but whether these younger patients are actually receiving their routine vaccinations in the pharmacy setting remains unclear. The objective of this study is to evaluate the availability of routine pediatric vaccination services in Washington State community pharmacies. 

## 2. Materials and Methods

Methods used in this study were determined to satisfy the criteria for Exempt Research by the Washington State University Human Research Protection Program (IRB#18151).

### Design

A 10-question phone survey was created to analyze pediatric vaccine availability and frequency of pediatric vaccine administration in the community pharmacy setting in Washington State. The selection of pharmacies to include in the survey was identified using the HealthMap Vaccine Finder tool [23]. The location parameters were set to be within a 50-mile radius of the following zip codes: 99202, 98926, 98101, and 99217. These specific zip codes were selected as they represent a variety of community pharmacies within Washington State that encompass both rural and urban communities. Pharmacy names were gathered from the search and contact information was then obtained from the website of each pharmacy. Only community pharmacies were included in the search in order to identify pharmacies where vaccines would likely be administered. Pharmacies located within hospitals and urgent care locations were excluded since in those settings it is more probable that any immunizations needed by the patients would be provided by someone other than a pharmacy team member, such as a nurse or medical assistant. A total of 67 pharmacies were identified to meet inclusion criteria and were subsequently contacted. 

A survey was designed with 10 questions to collect demographic information, availability of vaccines for pediatric patients, and pediatric vaccine administration frequency, as shown in Table 1. The survey was designed by a student pharmacist researcher and reviewed by a faculty member who oversees immunization content in the Doctor of Pharmacy program. Surveys were conducted by a student researcher under supervision of a pharmacist faculty member. The survey was conducted by telephone during normal business hours, and all pharmacy personnel (including pharmacists, pharmacy technicians, pharmacy interns or pharmacy assistants) were permitted to answer survey questions. Prior to collecting responses, the surveyor used a script to inform each participant of the purpose of the survey, that participation was voluntary, and that individually identifying information would not be collected. 

Pharmacy names were collected since the surveyor was contacting them directly, but individual names of participants were not recorded. In Microsoft Excel, each pharmacy name was removed, and pharmacies were thereafter referred to by number. Each survey took about approximately five to ten minutes to conduct with responses being written on paper then transcribed to Microsoft Excel. Researchers analyzed the vaccine availability by vaccine type, age requirements for administration, and patient request using descriptive statistics. 

During the survey, some responses included the use of synonymous vaccine names, which required researchers to categorize vaccines under one standard name. For the purposes of this study, vaccine results are referred to by the nomenclature used in the Recommended Child and Adolescent Immunization Schedule for ages 18 years or younger [4], as seen in Appendix A. However, sometimes other names were also reported by respondents. For example, if a pharmacy respondent reported offering “Prevnar”, that was counted as a response for “pneumococcal vaccine”. During data analysis, the term “pneumococcal vaccine” was used to include both pneumococcal conjugate vaccine (PCV-13) and pneumococcal polysaccharide vaccine (PPSV-23). PCV-13 is a 4-dose series recommended for all children, while PPSV-23 is recommended only for children in certain high-risk groups. When conducting the survey, the student researcher did not ask the pharmacy respondents to differentiate between the PPSV-23 and PCV-13 vaccines and instead categorized both as “pneumococcal vaccine”, so this terminology was utilized for analysis. Similarly, the term “tetanus, diphtheria, pertussis vaccine” was used during analysis to include both the DTaP vaccine, which is a 5-dose series for children below age seven, and the tetanus, diphtheria, pertussis (Tdap) vaccine, which is recommended for patients older than age seven who were not vaccinated with DTaP and also for adolescents starting at age 11. “Hepatitis vaccine” included vaccines against hepatitis A and B, which are offered as individual vaccines and as a combination product. Finally, “meningococcal vaccine” was the terminology used for analysis to include both vaccines for meningococcal serogroups A, C, W, and Y as well as meningococcal serogroup B vaccine. 

## 3. Results

Surveys were conducted between March 2020 and January 2021. Data collection began before the COVID-19 pandemic but was halted for six months following university policy for conducting research. Of the 67 pharmacies contacted, one was determined to be closed indefinitely, one was located within a medical care clinic and therefore did not meet study inclusion criteria, and personnel from three declined to participate. Personnel from the remaining 62 pharmacies were contacted and asked to participate in the survey. Four of the participating pharmacies (6%) reported that they did not offer pediatric vaccinations at all. Demographics for the respondents from the 58 pharmacies offering pediatric vaccinations are displayed in Table 2. 

Of the 58 pharmacies offering vaccinations for pediatric patients, the most commonly reported vaccines included influenza (97%) and tetanus, diphtheria, and pertussis (88%). Vaccines for hepatitis (71%) and human papillomavirus (HPV) (69%) were also commonly offered, followed by with meningococcal vaccines (66%), polio (45%), and Hib vaccine (40%). The results are presented in Figure 1. Nearly all (97%) pharmacies (56/58) surveyed reported having at least one of the ACIP-recommended routine childhood vaccinations available for patients under the age of 18 [4].

When asked how frequently respondents see pediatric patients (less than 18 years old) come in for vaccinations, 22 respondents (38%) reported that most of their pediatric patients are seen seasonally during influenza season or close to the start of the school year. Ten pharmacies (17%) reported administering pediatric vaccinations on a weekly basis, 13 pharmacies (22%) reported pediatric vaccinations on a monthly basis, and seven pharmacies (12%) reported that they very rarely see children come in for vaccines, if at all.

When asked if there are any vaccines that are requested by patients but are not at their pharmacy, several ACIP-recommended childhood vaccines were mentioned. Three pharmacies stated an unmet demand for MMR, two pharmacies had requests for HPV and hepatitis vaccines, and one pharmacy had a request for varicella. Other requested vaccines included other travel vaccines such as typhoid, rabies, cholera, and dengue, in addition to yellow fever. 

When asked what the minimum age for immunizations was per pharmacy, 16 pharmacies (28%) reported vaccinating patients two years and older and 20 pharmacies (34%) reported vaccinating patients three years and older. There was one pharmacy that was restricted to only immunizing pediatrics at least 13 years old, and the remaining pharmacies were able to vaccinate pediatrics four years and older (36%). Complete results are shown in Figure 2. 

## 4. Discussion

Concerns for declining pediatric immunization rates have increased since the beginning of the COVID-19 pandemic. In a large observational study of pediatric vaccination rates in US health systems, investigators report only 74% of infants under seven months were up to date on their vaccines in September of 2020 compared to 81% in the previous year, with the lowest rates seen in black infants [24]. Several other studies have also shown declining vaccine coverage for pediatrics. In Michigan, researchers report decreases in vaccine administration for patients under two years old in May of 2020 when compared to pre-pandemic data [25]. Evaluation of the Centers for Disease Control and Prevention’s (CDC’s) Vaccines for Children Program provider order data and the Vaccine Safety Datalink administration data also support substantial reductions in routine pediatric vaccine ordering and administration [7]. As COVID-19 restrictions continue to ease with many aspects of healthcare in the US returning to pre-pandemic levels [26] the decline in rates of routine pediatric vaccinations is not following the same trend [27]. In this study, we evaluated the availability of routine pediatric vaccination services in community pharmacies in Washington State. 

Although most pharmacies (97%) reported having at least one ACIP-recommended childhood vaccine in stock, all pharmacies described suboptimal administration of pediatric immunizations. Several pharmacies in this study report insurance barriers as a primary reason that they are unable to vaccinate some pediatric populations, since some Washington State plans require patients under the age of 18 to receive their vaccinations from their primary care provider’s office. In addition to limitations on insurance coverage, this disconnect may also be due to corporate or state policies restricting vaccination of certain age groups in the community pharmacy setting. The National Alliance of State Pharmacy Associations (NASPA) reports that although the PREP Act gives authority of pharmacy personnel to vaccinate patients over the age of three years, some states include specific variances to pharmacist immunization authority during a state-declared public health emergency [13]. NASPA reports that 22 states, including Washington State, allow pharmacists to administer all vaccines on the CDC-recommended immunization schedule for children as young as birth to six years old. Eighteen states restrict this age limit to seven years and older with several states requiring a doctor’s prescription [13]. Despite Washington State pharmacists being permitted to vaccinate children of any age with any CDC recommended vaccine, the youngest age group vaccinated by pharmacies in this study was 2 years and older with the majority of pharmacies reporting three years and older. This variance is likely common throughout the US where corporate policies may be more restrictive than state or federal authorizations.

Several discrepancies were noted during data collection in this study. Investigators noted variations in responses from pharmacy personnel of same chains but at different locations regarding the age of the patient that they were permitted to vaccinate as well as types of vaccines that location carried. This disparity could be due to differences in patient populations in the different communities that were surveyed. Rural areas may see more or fewer pediatric populations compared to urban settings, which would therefore affect the immunization inventory of that pharmacy. It is important to note that although several pharmacies reported a limited variety of on-hand vaccinations, these pharmacies did clarify that they are able to obtain any vaccine as requested by the patient or pursuant to a prescription. 

### Limitations

One important limitation to this study was the generalization of some survey results. When pharmacy personnel were asked “Which vaccines does your pharmacy offer for pediatric patients?” responses included phrases like “I can order anything”, “any vaccine available by prescription”, “the recommended vaccines”, or “routine vaccinations”. All these responses were counted as an equivalent response to “All vaccines are offered”. Due to these cumulative responses and the aggregation of data described in the methods, some immunization availability might be lower than reported in this study. 

Additionally, pharmacy personnel, which could include pharmacists, interns, technicians, and assistants, were allowed to complete the survey on behalf of the pharmacy location. Since the PREP acts require immunizing technicians and interns to complete substantial training, it is likely that technicians and interns who administer vaccinations were very knowledgeable about vaccines offered in their pharmacies. However, it is possible that technicians and interns who did not have experience with vaccinations, as well as pharmacy assistants who are not allowed to vaccinate, could be less informed on this topic. The variability in the roles and knowledge of the respondents could have led to some inaccurate responses.

The results of this study are restricted to pharmacies in Washington State and only in the community pharmacy setting. Washington State was selected as this is where the researchers were located, and the community setting was selected because of the known convenience and access for patients in these settings. Pharmacists can vaccinate patients in other healthcare settings such as ambulatory clinics and hospitals, and these data were not captured by this study. Since the surveys were conducted in different counties all throughout the state, we believe that these results are an accurate representation of general Washington State community pharmacy practice. 

The next steps for our research include investigation into the barriers of providing immunization services in the community pharmacy and expanding data collection to other states. Further research is also needed post-pandemic to evaluate how changing pharmacy practice models impact pediatric vaccination services. 

## 5. Conclusions

The declines in pediatric vaccination rates in the US secondary to the COVID-19 pandemic have not rebounded to pre-pandemic levels. This study described the availability of vaccine services among community pharmacies in Washington State. Nearly all respondent pharmacies reported having at least one of the ACIP-recommended routine childhood vaccinations available for patients under the age of 18. Community pharmacies are well-positioned to administer routine vaccinations to pediatric patients and address declining pediatric vaccination rates. Steps needed to address this disparity to improve coverage of routine childhood vaccinations include addressing corporate and state regulations on pharmacist-provided immunizations in the community setting, increasing awareness of these services for patients, and improving the engagement of pharmacies in taking on a larger role in the care of this population.

## Figures and Tables

**Figure 1 pharmacy-10-00156-f001:**
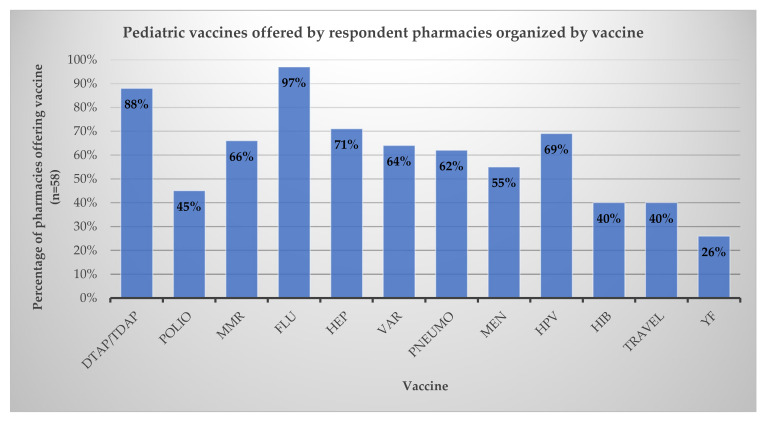
Pediatric vaccines offered in respondent pharmacies. Abbreviations used: DTAP—diphtheria, tetanus, acellular pertussis; TDAP—tetanus, diphtheria, acellular pertussis; MMR—measles, mumps, rubella; FLU—influenza; HEP: hepatitis; VAR—varicella; PNEUMO—pneumonia; MEN—meningitis; HPV—human papillomavirus; HIB—Haemophilus influenzae type b; TRAVEL—other travel vaccines; YF—yellow fever.

**Figure 2 pharmacy-10-00156-f002:**
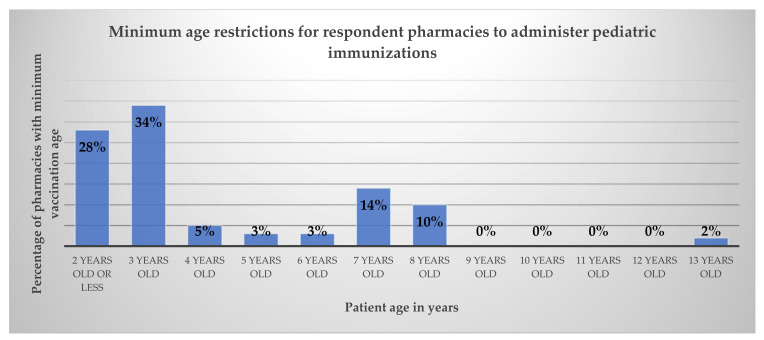
Minimum age required to administer pediatric immunizations as reported by pharmacy respondents.

**Table 1 pharmacy-10-00156-t001:** Survey questions.

Item Number	Question
1	What is your role in your current pharmacy position? (pharmacist, intern, technician, assistant)
2	What gender do you most identify with? (male, female, other, prefer not to answer)
3	What is your age?
4	Are you a full-time or part-time employee at your pharmacy?
5	How long have you worked in pharmacy practice?
6	How long have you worked at this pharmacy?
7	How often does your pharmacy immunize pediatric patients? (daily, weekly, monthly, yearly, or not at all)
8	What is the minimum age for patients to receive immunizations at your pharmacy? (age in years)
9	Which diseases does your pharmacy offer vaccines against for pediatric patients? (ACIP-recommended vaccines for pediatric patients)
10	Are there any pediatric vaccines requested by your current patients that you do not offer? (ACIP-recommended vaccines for pediatric patients)

Abbreviations used: ACIP: Advisory Committee on Immunization Practices. Possible response options included in parentheses when applicable.

**Table 2 pharmacy-10-00156-t002:** Respondent demographics.

Demographic	*n* (%)
Current pharmacy position	
Pharmacist	40 (69%)
Pharmacy Intern	8 (14%)
Pharmacy Technician	10 (17%)
Pharmacy Assistant	0 (0%)
Gender	
Male	17 (29%)
Female	41 (71%)
Prefer not to answer	0 (0%)
Employment status	
Full time	42 (72%)
Part time	16 (28%)
Age and Experience	
Average age	35.8 years (range 22 to 65 years)
Average years in practice	11.7 years (range <1 to 50 years)
Average years employed at this pharmacy	6.6 years (range <1 to 50 years)

## Data Availability

Study data will be made available upon reasonable request to corresponding author.

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
