# Peer review of "Examining Routine Pediatric Vaccination Availability in Community Pharmacies in Washington State"

_pharmacy, 2022, doi:10.3390/pharmacy10060156_

Round 1
Reviewer 1 Report
Thanks for the opportunity to review this manuscript. This brief report describes a telephone survey aimed at exploring the access to and availability of vaccination services for pediatric patients in the state of Washington. As the authors have noted, expanding vaccine availability for children through partnership with pharmacies is imperative to ensure an adequately immunized population and to promote community wellness. Overall this manuscript was clear and concise with a robust introduction. The robust survey response rate is a major strength of this manuscript.
Below are some comments that I believe would strengthen this manuscript.
1. Major comments
1.1. The title should be edited to be more descriptive of the paper. A better title should bring attention to the fact that this study examined pediatric vaccine services availability in community pharmacies within a specific geographic location.
1.2. Was the survey validated in any way, even if just face validation? This should be described in the methodology.
1.3. This survey seems to be subject to a fair amount of reporting bias considering that varying pharmacy staff members were able to participate. A technician may have a considerably different understanding of vaccine services in the pharmacy compared to a pharmacy manager, for example. To help the reader contextualize these data, I would suggest that the demographic parameters collected from the pharmacies be shared as a table in aggregate within the manuscript. Alternatively, a discussion of these parameters could be added to expand the results section. This would help the reader to make an informed assessment of the results.
1.4. Lines 269-271. I do not believe this study truly outlines, "... strengths and limitations..." It is more accurate to state that the study describes the availability of vaccine services among community pharmacies in Washington state.
2. Minor comments
2.1. Lines 17 and 45. Haemophilus is misspelled and should be capitalized.
2.2. Figure 2. To streamline this figure, I would use the CDC vaccine abbreviations on the x-axis rather than the whole name of the vaccine.
2.3. Figures 2 and 3. I would prefer to see these results presented in percentages.
2.4. Lines 206-208. I don't think this study evaluated the role of pharmacists. Instead, it focused on the availability and access of vaccines in community pharmacy settings. One question focused on the utilization of these services. Consider rewording to better reflect the data collected.
2.5. Lines 211-213. How were data collected on barriers to vaccination? I did not see this in the 10-question survey. Were these interviews semi-structured, or were there other questions asked that were not stated?
2.6. In the limitations, I would discuss the variety of respondents allowed to complete the survey, as this likely contributed to the variability in responses mentioned in the first paragraph of the limitations.
2.7. Lines 261-263. The response was robust (86.6%) for survey-based research. It is not apparent that workload and/or practice changes truly decreased the response rate. I would suggest deleting the comment regarding the response rate.
Author Response
Thank you for the helpful comments and feedback!

Reviewer 2 Report
Dear authors
The manuscript is well written and in a clear way.
The methods used allow you to achieve the proposed aim.
Very well described the inclusion and exclusion criteria.
Missed an Ethical Consent for this study.
The discussion needs to be deepened and supported by more references, only 8 are used.
Figures could be redesigned in a more scientific way. Also, figure legends should be more descriptive of Figure content.
Author Response

(The authors gave the same response as above.)

Reviewer 3 Report
Geographical reference (Washington state) must be included in the title and in the objective of the study
line 10: specify the acronym ACIP
line 40: Figure 1 is not readable, I suggest to delete or to move in Appendix in a bigger way
line 63: better specify in which groups
line 103: did the HealthMap Vaccine Finder tool include pharmacies that do not administered vaccines?
line 104: why do you select zip code instead of stratified by rural/urban area random selection. No analysis for rural / urban area is presented in the manuscript
line 112: a total of 67 out of how many?
line 151:question 1-6 are not relevant for the objective of the study also because in the analysis are included only pharmacies
questions 7-10 are not position-specific, why they are requested for any personnel? how do you manage if there are inconsistencies for the same pharmacy?
questions 8-10 are open question without possible response options?
To reach the study goal it would make sense to ask if the pharmacies started getting pediatric vaccinations during the pandemic or if they did before
line 159: question 7 seems to not include "never" as possible response (see also note to line 103)
Figure 2: add %!
Figure 3: add %
line 248: this is a phone survey that can be managed also in areas where the researchers were not located!
line 269-271: this study does not reach this goal
Author Response

(The authors gave the same response as above.)

Round 2
Reviewer 3 Report
None